



# Data rescue of daily climate station-based observations across Europe
**Joan Ramon Coll[1, *], Gerard van der Schrier[2], Enric Aguilar[1], Dubravka Rasol[3], Roberto**
**Coscarelli[4] and Andrés Bishop[1]**
[1] *Centre for Climate Change (C3), Rovira i Virgili University (URV), Vila-seca - 43480 Spain*
[2] *Royal Netherlands Meteorological Institute (KNMI), De Bilt – 3730 AE The Netherlands*
[3] *Croatian Meteorological and Hydrological Service (DHMZ), Zagreb - 10000 Croatia*
[4] *Consiglio Nazionale della Ricerche – Istituto di Ricerca per la Protezione Idrogeologica (CNR-IRPI),*
*Rende - 87036 Italy*
*Corresponding author: joanramon.coll@urv.cat*
Gerard van der Schrier: schrier@knmi.nl
Enric Aguilar: enric.aguilar@urv.cat
Dubravka Rasol: rasol@cirus.dhz.hr
Roberto Coscarelli: r.coscarelli@irpi.cnr.it
Andrés Bishop: andres.bishop@gmail.com





**ABSTRACT**

In the framework of the project *"Integrated approach for the development across Europe of user oriented climate indicators for GFCS high-priority sectors: agriculture, disaster risk reduction, energy, health, water and tourism"* (INDECIS 2017-2020), around 610K climate station-based observations were rescued over European regions for the main climate variables (maximum and minimum temperature, rainfall, sunshine duration and snow depth) along the 20th century at daily scale. Rescued data will constitute, together with other gathered regional datasets, the INDECIS-Raw-Dataset, which will expand current European data coverage contained in the European Climate Assessment & Dataset (ECA&D).

An extensive examination of the ECA&D dataset was conducted to find spatial-temporal data gaps or stations with low percentage of daily data as prior candidates for data recovery in European regions. This exercise led us to focus our efforts on the Central European region and the Balkans. Digitizing was carried out by using a rigorous "key as you see" method, meaning that the digitizers type the values provided by data images, rather than using any coding system. Digitizers carefully cross-checked the typed values against original sources for the 10th, 20th and 30th day of each month to make sure that no days were skipped or repeated during the digitizing process. Monthly totals and statistical summaries were computed from transcribed data and were compared with monthly totals and summaries provided by data sources to check accuracy as preliminary quality control. The digitizing method and the quality control of the digitizing process applied in this study ensured an accurate data transcription according to the obtained statistics.

The daily dataset rescued in this study across Europe is available at: https://doi.pangaea.de/10.1594/PANGAEA.896957


## 1. INTRODUCTION

Meteorological observations in machine readable format are necessary to study observed climate
variability and change and for the design of climate products and services, such as regional and global
climate models, among others. Nowadays, the lack of climate data for particular regions or for specific
historical periods is still affecting negatively climate products increasing the associated uncertainties
(Brunet and Jones, 2011). For this reason, data rescue missions are still necessary, especially in
developing countries and for pre-mid-20th century data since data stored in log-books or meteorological
notebooks are at risk to be lost (WMO, 2016).
Several efforts in the last two decades included data rescue missions in order to enhance the quality and
longevity of climate series and achieve a more accurate climate analysis. The European co-funded project
entitled "Uncertainties in Ensembles of Regional ReAnalyses" (UERRA 2014-2017) is perhaps one of
the most current projects which allocated a great human and economic resources for data rescue purposes.
UERRA project allowed to recover around 8.8M of synoptic meteorological observations of the Essential
Climate Variables (ECVs) across Europe and some regions of the Mediterranean basin for the period
1877-2012 (Ashcroft et al., 2018). The new high-quality UERRA dataset was submitted to the main global
and regional climate data repositories (e.g. Meteorological Archival and Retrieval System - MARS
Archive, European Climate Assessment and Dataset -ECA&D, International Surface Pressure Databank
- ISPD -, among others) with the aim to improve model outputs of regional reanalysis and estimate more
accurately the associated uncertainties.
On the other hand, the initiative undertaken by the Atmospheric Circulation Reconstructions over the
Earth (ACRE, Allan et al., 2011) is in charge to coordinate data rescue activities at global scale. Main
tasks are related with major data recovery, imaging and digitization of historical weather observations.
The "Mediterranean Data Rescue" (MEDARE) initiative and the "Historical Instrumental Climatological
Surface Time Series of The Greater Alpine Region" (HISTALP) are projects focused at regional scale
(Auer et al., 2007; Brunet et al., 2014a, 2014b). MEDARE, coordinated by WMO, aims to develop,
consolidate and progress climate data and metadata rescue activities across the Greater Mediterranean
Region. In the HISTALP project, leaded by the Central Institute of Meteorology and Geodynamics in
Austria (ZAMG), a regional database of monthly homogenized temperature, pressure, precipitation,
sunshine and cloudiness records was developed from rescued historical climate records. Other initiatives
are also carrying out at national scale leaded by National Meteorological and Hydrological Services
(NMHSs), such as in Germany (Kaspar et al., 2015).
The European co-funded project INDECIS (*Integrated approach for the development across Europe of*
*user oriented climate indicators for GFCS high-priority sectors: agriculture, disaster risk reduction,*
*energy, health, water and tourism*), leaded by the Rovira i Virgili University (Tarragona, Spain), will
develop user oriented climate indicators across Europe for the GFCS priority sectors (Water, Energy,



Health, Agriculture and Food Security, Disaster Risk Reduction) plus Tourism. The project includes
efforts in data rescue to expand current ECA&D dataset across the poorest climate data coverage over
some European regions. This paper presents this process. Station-based climate observations were rescued
over European sub-regions (mainly Central Europe and Balkans region) for the main climate variables
(maximum and minimum temperature, rainfall, sunshine duration and snow depth) along the 20th century
at daily scale. Rescued data will constitute, together with other gathered regional datasets, the newly
INDECIS-Raw-Dataset, which will expand current European data coverage included in the ECA&D
Dataset. INDECIS-Raw-Dataset will surely further improve the high quality climate products and
services across Europe.



## 2. MATERIALS AND METHODS

This section describes the resources and methodology used in this study to develop data rescue efforts undertaken in the framework of the INDECIS Project. The first step consisted of identifying data gaps in ECA&D dataset in order to flag the poorest covered regions across Europe. Once identified, the undigitized existing data sources for these particular regions were located and classified. Then, a digitization plan was designed by making an inventory of the priority meteorological stations/periods to be rescued. Climate data was digitized and the metadata for each meteorological station was collected and stored for future quality control and homogenization purposes. Finally, a preliminary assessment of data rescued was undertaken to visualize the added value of DARE efforts by identifying climate extreme events.

### 2.1. Inspection of data gaps in ECA&D dataset

Data rescue efforts were designed to improve spatial and temporal data coverage of the ECA&D dataset. The variables of interest were maximum and minimum temperature (TX/TN), rainfall (RR), sunshine duration (SS) and snow depth (SD) at daily scale.

An extensive examination of ECA&D dataset (http://eca.knmi.nl/) was conducted to find spatial and temporal data gaps across Europe. This preliminary exercise provided us valuable information about which European sub-regions presented lower density of stations (Fig. 1). Regions located in eastern Europe showed the lowest spatial climate data coverage and larger temporal data gaps. In particular, the Balkans region (Croatia, Republic of Serbia, Montenegro, Bosnia and Herzegovina and Republic of Macedonia) was identified as a key region for data rescue missions while other sub-regions from Central Europe (mainly Czech and Slovak Republics), the Mediterranean basin (Italy, Greece and Turkey) also showed a serious lack of climate data coverage. Otherwise, regions with highest density of climate series were focused mainly in Germany, Slovenia, Scandinavia, the Netherlands, Switzerland, France and Great Britain.

### 2.2. Identification of undigitized data sources

Once European sub-regions with lower availability of spatial and temporal climate data coverage were located, the data sources of undigitized records were identified for these particular sub-regions.
The Croatian Meteorological and Hydrological Service (DHMZ located in Zagreb, Croatia) responded positively to our request and provided  pdf files containing meteorological records directly scanned from original log-books.



In addition, other undigitized data sources were identified on-line thanks to the WMO MEDARE initiative
and the UERRA project through the United States of America's National Oceanic and Atmospheric
Administration/National Climatic Data Center (NOAA/NCDC) Climate Data Modernization Project
(CDMP: http://docs.lib.noaa.gov/rescue/data_rescue_home.html) for European eastern regions, the
Balkans and the Mediterranean basin (Ashcroft et al., 2018, Brunet et al., 2014a, 2014b). Synoptic station-
based observations of atmospheric pressure, air temperature, wind speed and wind direction were already
digitized at hourly scale under the UERRA project, but many other meteorological observations remained
undigitized at daily scale. The INDECIS project represented a great opportunity to rescue all this amount
of non-digitized daily data by using the same data sources already scanned.
Table 1 summarizes data sources obtained on-line through CDMP and also provided by the Croatian
Meteorological and Hydrological Service depending on each European sub-region and for different
periods along the 20[th] century and the first decade of the 21[st] century. All of these data sources were also
stored in a central server due to heavy size and to avoid data loses.
Most of data sources obtained on-line through CDMP were secondary. Unfortunately, secondary data
sources are more prominent to keep transcription errors than original data sources. Meteorological
observations were handwritten especially in early-20[th] century while they are typed since 1960s and 70s.
It is also worth to mention that the quality of scans was not always clear and readable and in some cases
the meteorological records were hard to read increasing the probability to make transcription errors when
digitizing.
Once data sources were thoroughly inspected, the digitization plan was designed taking into account the
spatial-temporal data gaps previously found in ECA&D dataset. Thus, an inventory of candidate climate
series to be rescued was created prioritizing those stations not included in ECA&D in order to increase
climate data spatial coverage across Europe. Those undigitized periods for the already existing stations at
ECA&D were also digitized to fill temporal data gaps, but not as a priority task.
A more detailed information about rescued climate series of the digitization plan can be found in Table
2, in which station metadata (e.g. station names, country, WMO code, latitude, longitude and altitude)
and type of variables digitised for each station are shown. Rescued periods were variable across time
covering the period 1949-2012 for the climate series located in the Balkans region and the period 1917-
1968 for climate series in Central Europe.
**2.3. Digitizing method**
Before starting with the digitizing procedure, a deep inspection of data sources was necessary to
familiarize with the general format, the structure of the data sheets and observations, the source language,
the measurement units and other additional notes which can provide valuable climate information

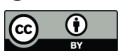


(metadata). This preliminary inspection of data sources is able to avoid gross digitizing errors derived
from some missing sheets for specific months, missing values or missing variables among others.
Figures 2, 3 and 4 show examples of the format and structure of scans obtained from various data sources.
In particular, Fig. 2, scanned from original log-books provided by the DHMZ, shows daily rainfall and
snow depth in Brodanci station (Croatia) due December 1983. It is a handwritten data source in which
meteorological records are combined with meteorological symbols and other notes for metadata storage
purposes. Figure 3, obtained on-line via CDMP, illustrates the structure of data sources for Central Europe
stations. The variables of interest were maximum and minimum temperature, rainfall and snow depth at
daily scale for Ceske Budejovice station (Czech Republic) due May 1960 in this case. Mainly typed values
are shown with some station identifiers as metadata. Figure 4, also obtained on-line via CDMP, shows
the structure and format of data sources for the Balkans region. The variables of interest were typed
maximum and minimum temperature, rainfall, snow depth and sunshine duration at daily scale for
Sarajevo station (Bosnia and Herzegovina) due July 1959.
Once all scans of data sources were thoroughly inspected, the digitizing process was set up. The digitizing
method used in this study consisted of applying a rigorous "key as you see" approach, meaning that each
digitizer was in charge to transcribe meteorological observations as were handwritten/typed in data
sources, without using any system code, following the recommendations given by WMO (2016).
Digitization was done over a spreadsheet designed for data insertion by following the format of each
variable in the hard copies. Half screen of the computer was used to read data from data sources and the
other half for typing meteorological records in the spreadsheet (Fig. 5). The digitizers used real-time
quality control strategies to minimize the instruction of erroneous values. They cross-checked the
digitized values against data sources every $10^{th}$, $20^{th}$ and $30^{th}$ for each month to check accuracy (to avoid
repeated or skipped values). Also compared monthly totals and averages of digitized values with the
monthly summaries provided in the hard copies, when they were available. Digitizing errors were reported
in a specific template (Fig. 6) while corrections were applied by using a copy of the first series to preserve
data traceability. The structure of the template used to document the preliminary quality control process
can be found in Fig. 6. This template informs us about some basic station metadata (e.g. country, name
of station and WMO/local code as identifier), the exact date and variable when a digitizing error was
produced (year, month, day and variable), the original value (erroneous) and the replacement value (the
correct one), the type of error (e.g. transcription error, source error, typing error…), the procedure applied
(corrected or set to missing) and any other comments for a better understanding of the type of error or the
final decision taken (e.g. hard to read, no sheet in data source, no station,…).
Obviously, this preliminary quality control was only applied to ensure that the digitizing procedure was
correctly carried out, but a second and more sophisticate layer of quality control routines must be run to
detect non-systematic errors hidden in climate data for future climate analysis (Aguilar et al., 2003;
Venema et al., 2012).



**2.4. Metadata collection**
Data gaps and potential unexpected variations in data sources were also recorded in a metadata
spreadsheet following the recommendations outlined by Aguilar et al. (2003).
Table 3 shows an example of a metadata template used for collecting additional notes for each station
divided in six basic sections.
The first section was designed to acquire metadata from data sources including the title of the source, the
period covered, the hosting, link (if any) to be found on-line and the variables. The second section was
related to station identifiers (stations name, country, WMO code, latitude, longitude and altitude (m))
while the third one contained valuable information about variables (variable name, units, period and
observing times). Section 4 was used to inform about special codes (e.g. code -99.9 for missing values,
or code -3 for rainfall < 0.1mm among others). Section 5 was used to describe the dates or periods with
missing values in data sources indicating the incident (e.g. no data for that station, hard to read values due
to poor quality of scans, or no sheet for any reason). Finally, section 6 was used to identify changes in
meteorological stations that could have an impact on observations, such as re-location of meteorological
station, instrumental changes, among others. This particular information is useful to understand
unexpected data behaviors or abrupt shifts for quality control and homogenization purposes.
**2.5. Computation of climate extreme indices**
Six of the 26 core climate extreme indices defined by the Expert Team on Climate Change Detection and
Indices (ETCCDI) (Peterson et al., 2001) plus two specific drought indices were selected to be computed
over Belgrade time-series for the whole period 1920-2017 to highlight, as example, the importance of
DARE efforts in terms of identifying climate extreme events. ETCCDI indices are based on daily
temperature values or daily precipitation amount. Some of them use fixed thresholds based on absolute
values meanwhile others use percentiles of the relevant data series to make comparisons between different
locations. The list of the 26 core ETCCDI indices and their definitions are available at:
http://etccdi.pacificclimate.org/list_27_indices.shtml. For this preliminary assessment six ETCCDI
indices were selected and computed at annual time scale to identify cold and dry years: TX10p, TN10p,
FD, CSDI, PRCPTOT and CDD. TX10p and TN10p indices shows the percentage of days when TX and
TN are lower than $10^{th}$ percentile (cold days and cold nights) computed for the base-period 1961-1990.
FD index reports the number of frost days (TN < 0ºC) per year meanwhile CSDI refers to the cold spell
duration index identifying the annual account of days with at least six consecutive days when TN < $10^{th}$
percentile. PRCPTOT index is the annual total precipitation in wet days and the CDD refers to the
maximum number of consecutive days with RR < 1 mm.



Two additional specific drought indices were also computed to identify major droughts in Belgrade series
for the period 1920-2017. The most widely used Standardized Precipitation Index (SPI) (McKee et al.,
1993) driven only by precipitation, and the Standardized Precipitation-Evapotranspiration Index (SPEI)
(Vicente-Serrano et al., 2010), based on the difference between the precipitation and the reference
evapotranspiration. Both drought indices were computed at the 6-month time scale to identify
accumulated dry conditions across time. Reference evapotranspiration were calculated by using the
Hargreaves algorithm (Hargreaves and Samani, 1985), which needs maximum and minimum temperature
together with extraterrestrial solar radiation (performed from latitude and the day of the year). The
calibration period was the longest period available for the Belgrade series (1920-2017) to compute both
SPI and SPEI indices following the recommendations outlined by Beguería et al., (2014) and Trenberth
et al., (2014).





**3. RESULTS**
This section describes the results derived from data rescue activities under the INDECIS project. After
applying the digitizing method detailed in section 2.3, the results in terms of amount of digitized values
and their spatial-temporal distribution are explained in this section. Results derived from the applied
quality control of the digitizing procedure are described and a preliminary assessment of the rescued data
is also carried out.
**3.1. Spatial-temporal distribution of rescued observations**
A total of 610K daily observations were rescued in the INDECIS project for maximum and minimum
temperature (in ºC), rainfall (in mm), sunshine duration (in hours) and snow depth (in cm) across Central
Europe and the Balkans region along really variable periods along the 20[th] century (Coll et al., 2019).
Figure 7 shows the spatial distribution of the 25 rescued climate series located in 7 European countries:
11 climate series in Czech Republic, 5 in Slovak Republic, 3 in Republic of Serbia, two in Bosnia and
Herzegovina, two more climate series in Republic of Macedonia, one in Croatia and the last one in
Montenegro (see also Table 4). The 25 climate series will be included in the INDECIS-Raw-Dataset
(together with gathered series obtained from other regionals datasets and not described in this study).
Table 4 shows a summary of number of rescued stations and total amount of digitised values for each
country. Rescued variables and periods are also described. Maximum and minimum temperature, rainfall
and snow depth were the rescued variables in Czech and Slovak Republic, while sunshine duration was
also included in the Balkans region (except in Croatia, where only rainfall and snow depth were digitized).
Digitizing periods were extended from 1917 to 1968 in Czech Republic and 1919-1968 in Slovak
Republic. In the Balkans region, digitizing periods focused on 1920-2012 in the Republic of Serbia, 1949-
1960 in Bosnia and Herzegovina, 1949-1984 in both Montenegro and Republic of Macedonia and, finally,
the period 1930-1990 was digitized in Croatia. Nevertheless, these common periods were really variable
among stations. More details about particular periods for each station can be found recovering Table 2.
Figures 8 and 9 show the total amount of digitized values for each country and for each variable,
respectively. The largest amount of digitized values corresponds to stations in the Czech Republic, which
nearly 250K values were rescued. Follow Slovak Republic with greater than 110K values, Republic of
Serbia with more than 85K values and Montenegro with nearly 65K values. Finally, the total amount of
digitized observations was lower in Croatia, Republic of Macedonia and in Bosnia and Herzegovina due
to the short length of digitizing periods, multiple data gaps and less variables to be digitized (e.g. in
Croatia).
A total of nearly 260K values were rescued in both maximum (TX) and minimum (TN) temperature (Fig.
9) meanwhile close to 160K values and greater than 150K values were rescued related to rainfall (RR)



and snow depth (SD), respectively. In less proportion, greater than 40K values were rescued related to
sunshine duration (SS). The main differences among the amount of digitized values for each variable
depended basically on the availability (or not) of such variables in the data sources.
**3.2. Quality Control**
The quality control of the digitizing process was applied to all climate series rescued in the INDECIS
Project. Monthly totals and sums provided by data sources (in most of the cases) were accurately cross-
checked with monthly totals and sums computed from digitized data. Results demonstrated that the errors
occurred during the digitizing process represented only the 0,6% of the total amount of digitized values,
which highlights the accuracy and high standards of the process and ensures the transmission of ready-
to-use data series. Most of errors occurred due to hard to read records (around 76% of errors; Fig. 10).
The main cause was the low quality of particular sheets in the scanned data sources. The second cause of
errors was variable confusion or, what means the same, column confusion in data sources (around 17%
of errors). In those cases, the digitizer did not realize that they were typing the wrong variable. This could
be solved by using templates that exactly match data sources with spreadsheets. Finally, the 7% of errors
were typing errors produced during the digitizing process (e.g. type 104,5 ℃ instead 10,5 ℃). All errors
found in the preliminary quality control of the digitizing process were successfully corrected or were set
to missing in the cases that a new value could not be offered.
The 25 new climate series will be incorporated to the INDECIS-Raw-Dataset and submitted for addition
into the ECA&D Dataset. Thus, the spatial-temporal climate coverage will surely improve in Central
Europe and in the Balkans region. More exhaustive quality control routines are strongly recommended to
find non-systematic errors together with the application of some homogenization tests to ensure the high
quality of the new dataset to be used for future climate analysis.
**3.3. Preliminary assessment of rescued data**
In this section, we intend to visualize the effects of data rescue and explain the impact of data rescue over
climate series. No need to say that the solid climatological conclusions cannot be drawn from them, as
the data has not been assessed for homogeneity, but the benefits of data rescue are highlighted.
For example, Fig. 11 shows the evolution of daily maximum (TX) minimum (TN) temperature and
precipitation (RR) at Belgrade station (Republic of Serbia) for the period 1920-2017. Data rescue efforts
allowed to extend 15 years back to 1936 creating a long-term time series of almost 100 years of records.
Focusing on the rescued period (blue line), extreme cold temperatures can be identified in 1922, 1935,
but especially in 1929. In particular, February 1929 was extremely cold in Belgrade reaching temperatures
on record in both maximum and minimum temperature for the whole time series. Minimum temperature

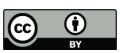



reached -25,5ºC and maximum temperature did not exceed -18,5ºC in a particular day. The evolution of
precipitation (Fig. 11) for the rescued period 1920-1935 shows dry conditions in 1920-1921, 1923 and
1928 meanwhile wet conditions were predominant in 1924-1927 and 1931-1933.
Data rescue efforts extend climatological analysis to the past. Even though the Belgrade series shown in
Fig. 11 are neither quality controlled and homogenized, the calculation of some ETCCDI indices
(Peterson et al., 2001) plus two specific drought indices (SPI and SPEI) suggests some climate features
that could not be studied before this DARE effort. For example, in the cold 1929 year (Fig. 12) or the dry
event experienced in 1920-1921 in Belgrade (Fig. 13).
According to these indices (Fig. 12) 1929 is identified as a cold year, with a high percentage (> 20%) of
cold days (TX10p) and nights (TN10p), over 100 frost days (FD) and 55 days singled out as part of a cold
spell (CSDI). The mentioned cold spell occurred in February 1929 and was general over most of Europe
being the coldest month on record in Poland (Sirocko et al., 2012). The Rhine river was frozen in Germany
taking into account that only occurred it six times during the 20$^{th}$ century and the canals were also frozen
in Venice according to the Meteorological Magazine published for the UK Meteorological Office in
March 1929.
Figure 13 shows the PRCPTOT, SPI 6-month, SPEI 6-month and CDD indices computed over the
Belgrade rescued time series for the period 1920-2017. These specific extreme indices were selected to
identify the dry event occurred in 1920-1921. Annual precipitation amount (PRCPTOT index) was low
in 1920 compared with other years of the time series (< 500 mm) and the consecutive dry days index
(CDD) shows that there was a period with more than 40 days with precipitation less than 1 mm. The
computation of additional drought indices such as SPI 6-month and SPEI 6-month allowed to identify the
driest event of the whole Belgrade series reaching maximum severity in 1921. This drought event not
only affected a particular European region, but most of European countries suffered severe dry conditions
during several months between years 1920 and 1923 (Hanel et al., 2018). In fact, West Europe was in
serious drought during 1920 and 1921, which was reported by the "Townsville Daily Bulletin" in July
1921. High pressure systems from the Azores remained stuck for almost the entire year, leading to clear
skies and dire shortages of rain. Most rivers in France were below the lowest records in 50 years, the
mountain torrents in Switzerland were not a third of their usual volume and the dry sequence lasted 86
consecutive days for most of Britain.

**4. DATA AVAILABILITY**

The daily dataset rescued in this study across Europe is available at the PANGAEA repository:
https://doi.pangaea.de/10.1594/PANGAEA.896957






## 5. SUMMARY AND CONCLUSIONS

In the framework of the INDECIS Project, some human and economic resources were allocated for data rescue activities across Europe in order to enhance the quality in the already existing climate products and services. This study deeply describes all the process carried out: from the identification of data gaps in ECA&D dataset and the inspection of undigitized data sources to the digitizing process together with the accurate documentation of data and metadata, including also the corrections derived from digitizing errors.

The process of identifying data gaps, the inspection of data sources to be rescued and the preparation of aforementioned data sources was actually a time consuming task (Bröninnmann, 2006). In particular, several hours of work and human resources were needed during the manual-keying digitizing process. For this reason, it was crucial to design and implement an effective and reliable digitizing method to obtain the final high-quality climate dataset avoiding extra-costs.

Some recommendations are available to guide experts involved in data rescue projects or initiatives. In this line, Bröninnmann (2006) designed a digitizing guide for climate data describing the use of technologies based on optical character recognition (OCR) technologies or based on speech recognition techniques to be faster in the digitizing procedure. Nevertheless, the study demonstrated that the manual-keying digitizing process was the most efficient method in terms of agility, reduction of transcription errors and post-process time consuming. The World Meteorological Organization supported this statement (WMO, 2016) recommending the use of OCRs only in a certain data sources, since human eye is still more effective transcribing handwritten data sources.

Nowadays, the most effective method of digitization is double or triple-keying data by using templates that match with format of original data sources (WMO, 2016). Despite this, the final economic cost is remarkably higher and most of projects cannot assume this extra cost. Simple manual-keying with an effective quality control during and at the end of the digitization process resulted the better balance between costs and data quality of rescued datasets knowing that some issues to solve already exist (Ashcroft et al., 2018).

In summary, a total of 25 climate series (610K daily observations) were rescued in this study for 7 countries of the Central Europe and the Balkans region along the 20[th] century by using the manual-keying digitizing method together with a preliminary quality control of the digitizing procedure (Coll et al., 2019). Climate variables of interest were maximum and minimum temperature, rainfall, sunshine duration and snow depth. The aforementioned rescued climate series will be included in the newly INDECIS-Raw-Dataset, which will be automatically ingested by the ECA&D Dataset to fill the spatial-temporal data gaps previously identified across Europe.

Rescued dataset will be submitted to more rigorous quality control routines to detect non-systematic errors (Aguilar et al., 2003) together with some homogenisation tests (Venema et al., 2012) to ensure a high-



quality and homogeneous data to be used by the international research community to design and
implement new climate products and services.
Future European climate analysis will be benefited of DARE efforts undertaken in this study such as
increasing the reliability of long-term climate trends or identifying historical climate extremes among
others.

**6. AUTHOR CONTRIBUTION**

**Joan Ramon Coll**: Searcher of undigitised data sources, developer of data inventories, in charge of the
manual digitization process (typing), extreme indices computation and analysis and manuscript
preparation.
**Gerard van der Schrier**: Everything related to ECA&D management: Inventory of digitised
stations/periods, provider of digitised data and ECA&D data gaps inspection).
**Enric Aguilar**: Designer of the digitization plan, supervisor of the digitization process and quality control
and paper structure designer.
**Dubravka Rasol**: Scanning and providing undigitized data for the Balkan region.
**Roberto Coscarelli**: Supervisor of the extreme indices analysis and detection of extreme events and also
paper reviewer.
**Andrés Bishop**: In charge of quality control process of digitization.

**ACKNOWLEDGEMENTS**

The Project INDECIS is part of ERA4CS, an ERA-NET initiated by JPI Climate, and funded by
FORMAS (SE), DLR (DE), BMWFW (AT), IFD (DK), MINECO (ES), ANR (FR) with co-funding by
the European Union (Grant 690462).

The authors declare that they have no conflict of interest.

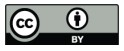



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





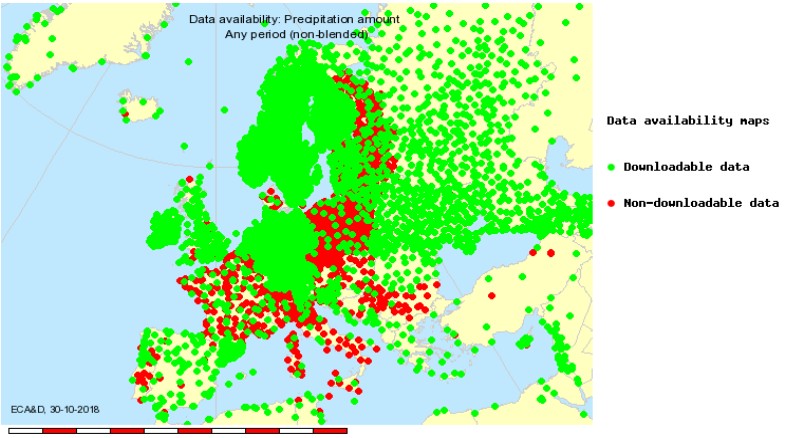

**Fig. 1**: Spatial distribution of meteorological stations in ECA&D (precipitation as example) across
503          Europe in 2018. Downloadable stations are in green and non-downloadable stations in red.



**Fig. 2**: Structure of original log-books (scans) provided by the Croatian Meteorological Service;
507                    Brodanci station (Croatia), December 1983.






**Fig. 3**: Structure of data sources (scans) for Central Europe stations: Ceske Budejovice station (Czech
511                                       Republic), May 1960.


**Fig. 4**: Structure of data sources (scans) for the Balkans region: Sarajevo station (Bosnia &
514                                       Herzegovina), July 1959.


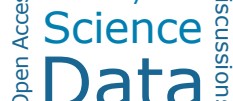



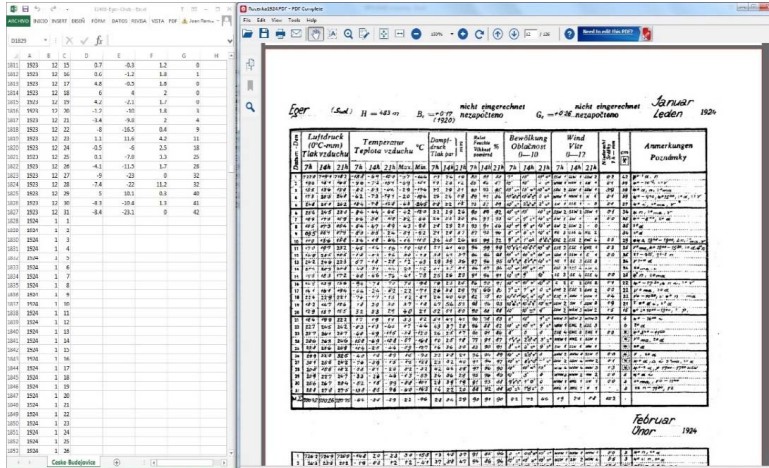


**Fig. 5**: Example of the manual-keying data transcription method used during the digitization process;

518       from scanned data sources (right) to digital spreadsheets (left).


| | A | B | C | D | E | F | G | H | I | J | K | L | M | N |
|---|---|---|---|---|---|---|---|---|---|---|---|---|---|---|
| 1 | country | station name | code | Year | Month | Day | Element | Original value | Replacement value | Detection test | Type of error | Procedure | Comments | |
| 2 | Croatia | Brodanci | 5080 | 1931 | 2 | 27 | RR | 0 | 0,2 | Visual checking | Transcription | corrected | Dificulties to be read | |
| 3 | Croatia | Brodanci | 5080 | 1931 | 2 | 28 | RR | 0 | 4,6 | Visual checking | Transcription | corrected | Dificulties to be read | |
| 4 | Croatia | Brodanci | 5080 | 1931 | 10 | 21 | RR | 4 | 4,9 | Visual checking | Transcription | corrected | Dificulties to be read | |
| 5 | Croatia | Brodanci | 5080 | 1931 | 10 | 27 | RR | 3 | 3,9 | Visual checking | Transcription | corrected | Dificulties to be read | |
| 6 | Croatia | Brodanci | 5080 | 1931 | 12 | 10 | RR | 1 | 0,9 | Visual checking | Transcription | corrected | Dificulties to be read | |
| 7 | Croatia | Brodanci | 5080 | 1933 | 1 | 25 | RR | 3,4 | 3,7 | Visual checking | Transcription | corrected | Dificulties to be read | |
| 8 | Croatia | Brodanci | 5080 | 1933 | 2 | 23 | RR | 5,9 | 5,1 | Visual checking | Transcription | corrected | Dificulties to be read | |
| 9 | Croatia | Brodanci | 5080 | 1933 | 3 | 7 | RR | 0,2 | 0,6 | Visual checking | Transcription | corrected | Dificulties to be read | |
| 10 | Croatia | Brodanci | 5080 | 1933 | 6 | 29 | RR | 2,5 | 2,8 | Visual checking | Transcription | corrected | Dificulties to be read | |
| 11 | Croatia | Brodanci | 5080 | 1933 | 8 | 2 | RR | 1,5 | 1,6 | Visual checking | Transcription | corrected | Dificulties to be read | |
| 12 | Croatia | Brodanci | 5080 | 1933 | 10 | 14 | RR | 0,9 | 6,9 | Visual checking | Transcription | corrected | Dificulties to be read | |
| 13 | Croatia | Brodanci | 5080 | 1934 | 6 | 12 | RR | 4 | 4,9 | Visual checking | Transcription | corrected | Dificulties to be read | |
| 14 | Croatia | Brodanci | 5080 | 1936 | 5 | 27 | RR | 2,2 | 12,2 | Visual checking | Source error | corrected | Typing error | |
| 15 | Croatia | Brodanci | 5080 | 1936 | 7 | 4 | RR | 37,3 | 37,1 | Visual checking | Transcription | corrected | Dificulties to be read | |
| 16 | Croatia | Brodanci | 5080 | 1936 | 10 | 2 | RR | 10,8 | 10,3 | Visual checking | Transcription | corrected | Dificulties to be read | |
| 17 | Croatia | Brodanci | 5080 | 1937 | 2 | 18 | RR | 0,8 | 0,07 | Visual checking | Transcription | corrected | Dificulties to be read | |
| 18 | Croatia | Brodanci | 5080 | 1938 | 11 | 1 | RR | 3 | 3,3 | Visual checking | Transcription | corrected | Dificulties to be read | |
| 19 | Croatia | Brodanci | 5080 | 1939 | 6 | 28 | RR | 19,8 | 19,3 | Visual checking | Transcription | corrected | Dificulties to be read | |
| 20 | Croatia | Brodanci | 5080 | 1939 | 9 | 22 | RR | 2,6 | 9,2 | Visual checking | Transcription | corrected | Typing error | |
| 21 | Croatia | Brodanci | 5080 | 1939 | 10 | 28 | RR | 2,5 | 2,8 | Visual checking | Transcription | corrected | Dificulties to be read | |
| 22 | Croatia | Brodanci | 5080 | 1940 | 2 | 1 | RR | 2,8 | 2,9 | Visual checking | Transcription | corrected | Dificulties to be read | |


**Fig. 6**: Template used to report the quality control of the digitization process.





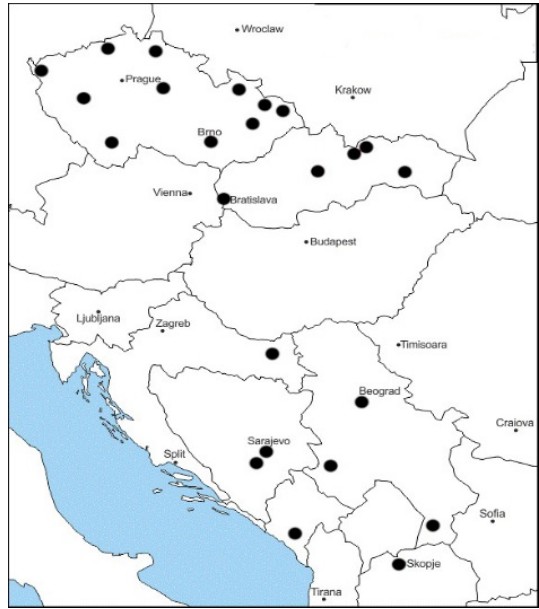

**Fig. 7**: Spatial distribution of rescued stations

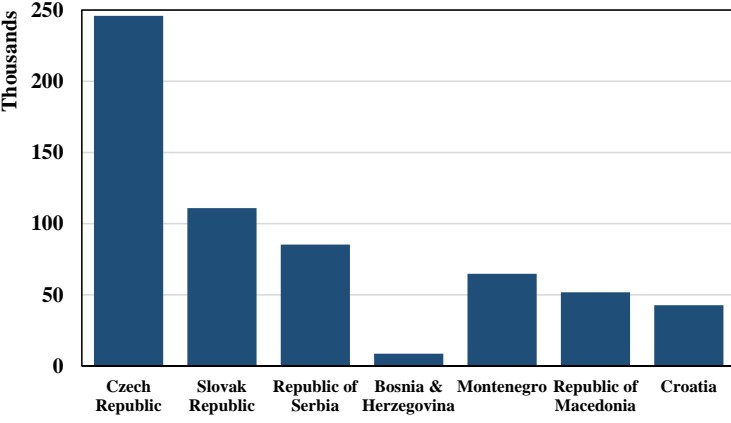

**Fig. 8**: Total amount of digitized values (in thousands) by countries



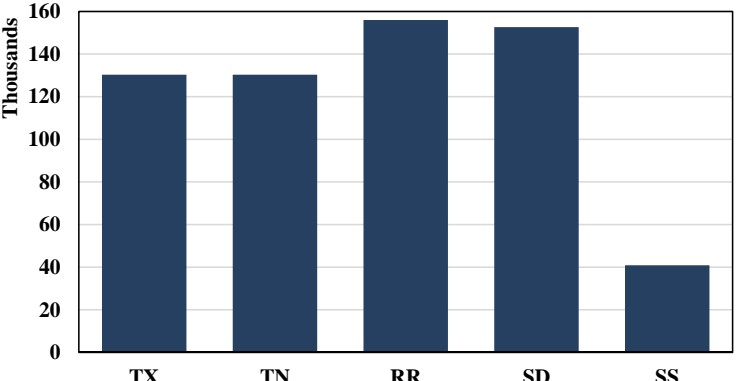

**Fig. 9**: Total amount of digitized values (in thousands) by variables


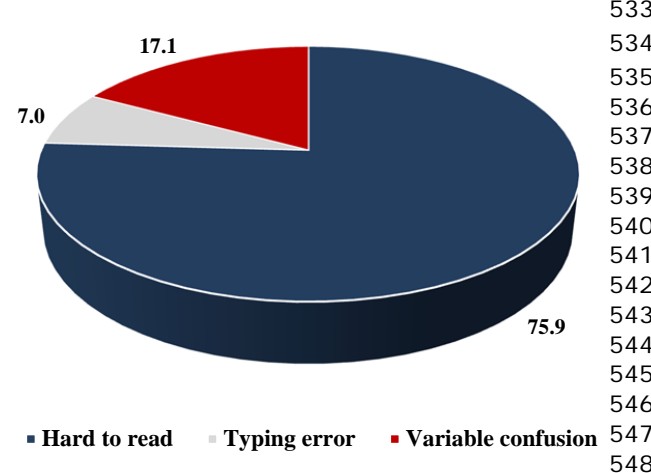


**Figure 10**: Percentage of type of errors found after the quality control of the digitizing procedure.

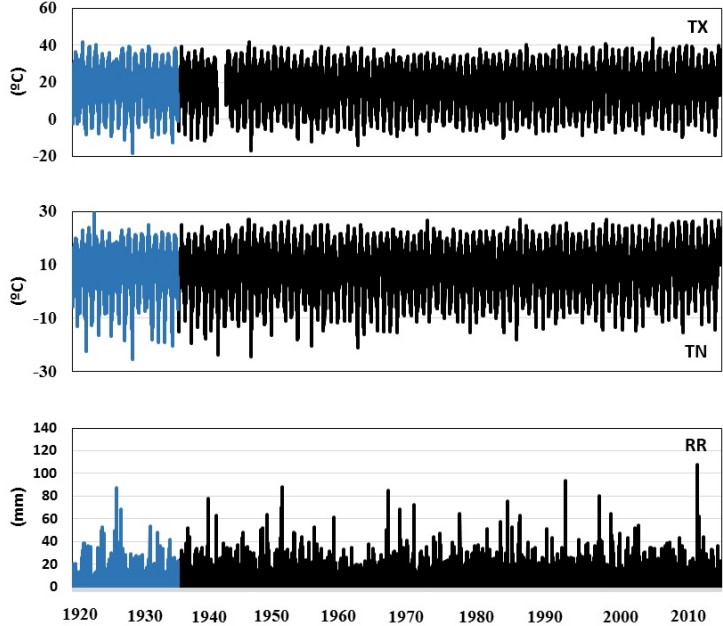

**Figure 11**: Evolution of daily maximum (TX) minimum (TN) temperature and precipitation (RR) at
Belgrade station (Republic of Serbia) for the period 1920-2017. The period 1920-1935 was rescued in
this study (blue line) meanwhile the period 1936-2017 was obtained from ECA&D Dataset (dark line).

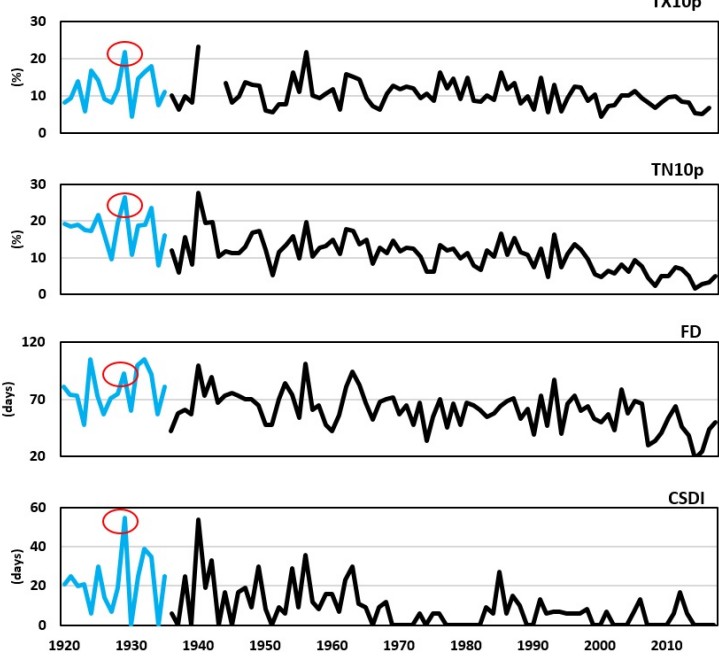

**Figure 12**: Time series of TX10p, TN10p, FD and CSDI extreme indices at Belgrade station (Republic
of Serbia) for the period 1920-2017. The period 1920-1935 was rescued in this study (blue line)
meanwhile the period 1936-2017 was obtained from ECA&D Dataset (dark line). Red circle shows a
climatic extreme (cold year) identified in 1929.

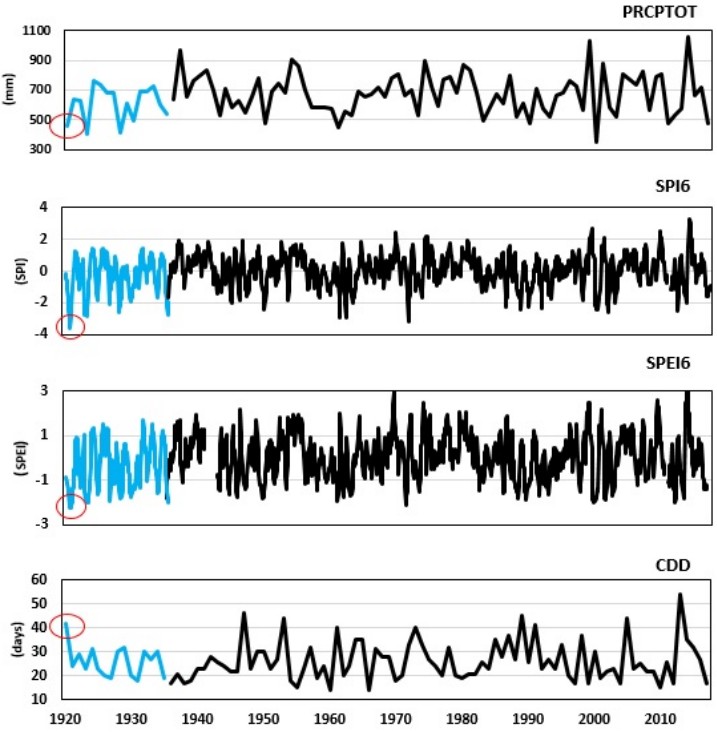

**Figure 13**: Time series of PRCPTOT, SPI 6-month, SPEI 6-month and CDD extreme indices at Belgrade
station (Republic of Serbia) for the period 1920-2017. The period 1920-1935 was rescued in this study
(blue line) meanwhile the period 1936-2017 was obtained from ECA&D Dataset (dark line). Red circle
shows a climatic extreme (dry years) identified in 1920-1921.

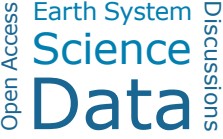

**Table 1:** Documental data sources used for data rescue purposes.

| Region | Documental Source | Period |
|---|---|---|
| **Central Europe** | Rocenka povetrnostnich posoro vani site statniho ustavu meteorologickeho. | 1916-1946 |
| | Rocenka povetrnostnich pozorovani meteorologickeho stanie Republiky Ceskoslovenshe. | 1948-1968 |
| | Rocenka povetrnostnych pozorovani observtoria na Lomnickom Stite. | 1940-1974 |
| **Balkans Region** | Izvestaj meteoroloske opservatorije u Beogradu. | 1920-1945 |
| | Resultati osmatranija u Beogradu. | 1946-1950 |
| | Meteoroloski godisnjak. I. | 1949-2012 |
| | Scans from original log-books provided by the Croatian Meteorological and Hydrological Service (DHMZ) | 1930-1990 |






**Table 2**: Rescued data included in the INDECIS-Raw-Dataset. Specific metadata such as country, WMO
code, station name, latitude, longitude, altitude and digitizing period are also shown. Digitized variables
are maximum (TX) and minimum (TN) temperature, rainfall (RR), snow depth (SD) and sunshine
duration (SS).

| Country | WMO code | Station Name | Lat. N | Lon. E | Alt. (m) | Variables | Digitizing period |
|---|---|---|---|---|---|---|---|
| **Czech Republic** | 11542 | Ceske Budejovice | 48°58'00" | 14°28'00" | 389 | TX/TN/RR/SD | 1917-1938 |
| | 11748 | Prerov | 49°28'00" | 17°27'00" | 214 | TX/TN/RR/SD | 1917-1952 |
| | 11406 | Eger/Cheb | 50°05'00" | 12°24'00" | 483 | TX/TN/RR/SD | 1919-1936 |
| | 11763 | Troppau/ Opava | 49°56'00' | 17°53'00" | 268 | TX/TN/RR/SD | 1917-1937 |
| | 11461 | Teplitz-Schnonau | 50°39'00" | 13°48'00" | 229 | TX/TN/RR/SD | 1917-1936 |
| | 11446 | Plzen | 49°44'00" | 13°80'00" | 357 | TX/TN/RR/SD | 1948-1953 |
| | 99999 | Turnov | 50°36'00" | 15°10'00" | 280 | TX/TN/RR/SD | 1948-1951 |
| | 11721 | Brno-Kvetna | 49°12'00" | 16°34'00" | 233 | TX/TN/RR/SD | 1948-1968 |
| | 11735 | Praded | 50°05'00" | 17°14'00" | 1490 | TX/TN/RR/SD | 1948-1957 |
| | 11622 | Caslav-Filipor | 49°54'00" | 15°24'00" | 252 | TX/TN/RR/SD | 1946-1960 |
| | 99999 | Frycovice | 49°41'00" | 18°13'00" | 274 | TX/TN/RR/SD | 1946-1953 |
| **Slovak Republic** | 99999 | O.-Gyalla/ Stara Dala | 47°53'00" | 18°12'00" | 120 | TX/TN/RR/SD | 1919-1937 |
| | 99999 | St. Smokovec | 49°08'00" | 20°13'00" | 1018 | TX/TN/RR/SD | 1921-1937 |
| | 11814 | Bratislava-Trnavaka | 48°10'00" | 17°08'00" | 139 | TX/TN/RR/SD | 1946-1968 |
| | 11931 | Skalnate Pleso | 49°12'00" | 20°55'00" | 1778 | TX/TN/RR/SD | 1946-1960 |
| | 11968 | Kosice | 48°42'00" | 21°16'00" | 206 | TX/TN/RR/SD | 1946-1950 |
| **Republic of Serbia** | 13274 | Belgrade | 44°48'00" | 20°28'00" | 132 | TX/TN/RR | 1920-1935 |
| | 13367 | Zlatibor | 43°44'00" | 19°43'00" | 1028 | TX/TN/RR/SD/SS | 1992-2012 |
| | 13489 | Vranje | 42°33'00" | 21°55'00" | 432 | TX/TN/RR/SD/SS | 1999-2012 |
| **Bosnia & Herzegovina** | 13353 | Sarajevo | 43°52'00" | 18°26'00" | 630 | SD/SS | 1949-1960 |
| | 13352 | Bjelasnica | 43°43'00" | 18°16'00" | 2067 | SD/SS | 1953-1960 |
| **Montenegro** | 13462 | Titograd/ Podgorica | 42°26'00" | 19°17'00" | 52 | TX/TN/RR/SD/SS | 1949-1984 |
| **Republic of Macedonia** | 13491 | Skopje | 41°59'00" | 21°28'00" | 240 | TX/TN/RR/SD/SS | 1949-1972 |
| | 13586 | Skopje (Petrovac) | 41°58'00" | 21°39'00" | 238 | RR/SD/SS | 1974-1984 |
| **Croatia** | 5080 | Brodanci | 45°32'33" | 18°27'26" | 92 | RR/SD | 1930-1990 |






**Table 3**: Metadata collection by using specific templates

| Metadata on data sources | | | |
|---|---|---|---|
| *Title of the source:* Meteoroloski godisnjak. I | | | |
| *Period covered by the source:* 1949-1978 | | | |
| *Available at:* CDMP-NOAA: http://library.noaa.gov/Collections/Digital-Documents/Foreign-Climate-Data-Home | | | |
| *Variables included:* Maximum and minimum temperature, rainfall and snow depth | | | |
| **Station Identifiers** | | | |
| *Station Name:* | Ceske Budejovice | *WMO code:* | 11542 |
| *Country:* | Czech Republic | *Altitude (m):* | 389 |
| *Latitude:* | 48º58'00" | *Longitude:* | 14º28'00" |
| **Variables Metadata** | | | |
| *Variable* | *Units* | *Period* | *Observing times* |
| Max. Temperature (TX) | (ºC) | 1917-1938 | Daily |
| Min. Temperature (TN) | (ºC) | 1917-1938 | Daily |
| Rainfall (RR) | (mm) | 1917-1938 | 7am |
| Snow depth (SD) | (cm) | 1917-1938 | 7am |
| **Special Codes** | | | |
| *Variable* | *Code* | *Description* | |
| TX/TN/RR/SD | -99,9 | Missing value | |
| Rainfall | -3 | Rainfall < 0.1mm | |
| Rainfall | -4 | Cumulative precipitation | |
| Snow depth | 0,1 | Snow traces on the soil | |
| **Missing values and/or periods** | | | |
| *Dates/Periods* | | *Incident* | |
| from 01/01/1919 to 31/07/1919 | | *No data* | |
| from 16/02/1921 to 31/05/1921 | | *Hard to read* | |
| from 25/03/1928 to 31/03/1928 | | *Hard to read* | |
| from 01/02/1931 to 31/03/1931 | | *No data* | |
| **Station Metadata (if available)** | | | |
| *Period of the incidence* | | *Type of incidence* | |
| December 1929 | | Instrument changes: Thermometer | |




**Table 4**: Summary of number of rescued stations and total amount of digitized values for each country
and period. Variables are maximum (TX) and minimum (TN) temperature, rainfall (RR), snow depth
(SD) and sunshine duration (SS).

| Country | Nº stations | Variables | Period | Total digitized | % |
|---|---|---|---|---|---|
| Czech Republic | 11 | TX/TN/RR/SD | 1917-1968 | 245935 | 40,3 |
| Slovak Republic | 5 | TX/TN/RR/SD | 1919-1968 | 110873 | 18,2 |
| Republic of Serbia | 4 | TX/TN/RR/SD/SS | 1920-2012 | 85343 | 14,0 |
| Bosnia & Herzegovina | 1 | TX/TN/RR/SD/SS | 1949-1960 | 8642 | 1,4 |
| Montenegro | 1 | TX/TN/RR/SD/SS | 1949-1984 | 64816 | 10,6 |
| Republic of Macedonia | 2 | TX/TN/RR/SD/SS | 1949-1984 | 51836 | 8,5 |
| Croatia | 1 | RR/SD | 1930-1990 | 42709 | 7,0 |
