# Peer review of "Data rescue of daily climate station-based observations across Europe"

_Earth System Science Data, 2019_

## Referee Comment (RC1) · Anonymous Referee #1 · 27 Mar 2019

The paper describes the digitisation and examination of daily climate data (610K observations of maximum and minimum temperature, rainfall, sunshine duration, snow depth) that have been digitised in the framework of a European project and which will be included in the ECA&D data set. The paper accompanies a publication of the data on PANGEA. This falls within the scope of the journal. The paper is short (well, it is not a huge data set), focused on the digitising process, and generally well written. However, I have a few comments that I would like to authors to address.

General: The English is sometimes a bit awkward - please have your manuscript checked by a native speaker.

General: I found the project in I-DARE (https://www.idare-portal.org/), but unfortunately not in the State of Data Rescue Assessment

(https://rmets.onlinelibrary.wiley.com/doi/10.1002/gdj3.56)

Data on PANGEAE: I did not find a legend for the "Reference" column. What does so33 mean?

Abstract: Please mention somewhere the time period covered.

Abstract: Can you say something on metadata in the abstract? The systematic collection with the form given in the paper is worth mentioning.

Introduction: Perhaps the paper could cite Thorne et al. (BAMS, 2017) as a vision where all global terrestrial data set eventually could end up.

L. 151: No pressure measurements were digitised, although they appear on the form. It is not clear whether this has already been done (it is somehow implied, but not explicit). Pressure is an important variable. Even though it is not stated in the manuscript as a focus variable, it would still have been good to collect it along with the other data.

The abstract mentions "preliminary quality control". I find this very useful. However, in my view this is not sufficient for calculating extreme indices. The QC is focused only on digitising errors, not on many other possible error sources. Also, no break-point detection is applied. However, extreme indices may be sensitive to outliers as well as to biases. The authors are clearly aware of that and mention it on lines 301 and following, but then they still go on. I know that this is only for illustration of what the data could deliver, what they add to the already existing material. But perhaps this should be phrased even more carefully.

Table 1: It would be good to have more information such as number of stations, provenance (repository/archive), perhaps number of pages/images.

---

## Referee Comment (RC2) · Anonymous Referee #2 · 21 Apr 2019

Review ESSD-2019-0, Rescue daily climate data

Regardless of its political wrappings (GFCS, WMO, etc.) does this data product add substantial reliable value to European climate data? If so, how can users verify? What have the authors accomplished by their efforts that make this data rescue effort notable and that qualifies this product for publication in ESSD? Clear answers do not emerge from the present manuscript.

Page 5 line 100: Authors introduce the acronym "DARE" which this reviewer understands as shorthand for DAta REscue but we get no definition. The authors use this term several times throughout the manuscript so we need a definition.

Figure 1 highlights two potential data deficiencies: absence of stations/data (beige background color) and existence of stations but data not downloadable (red dot). But the narrative talks about spatial data coverage and temporal data gaps, while the rescue products refer to undigitized data. In fact Spain - with relatively weak spatial distribution at least according to Fig 1 - and even France have substantial spatial data coverage problems but apparently nothing to rescue?  But didn't France already address their data coverage issues through their SCOPE climate project, e.g. https://doi.org/10.5194/essd-11-241-2019? So if this study then focused on the red dot regions, e.g. where data exist but either not downloadable (according to caption in Figure 1) or not digitized (according to narrative) - and the reader gets no overt explanation of difference if any between not downloadable and not digitized - then northern Europe particularly Poland and Czechia seem most 'red'. If the Balkans represent a "a key region for data rescue missions" (page 5 line 113) that designation must include additional factors such as relative ease of access to non-digital data? Saying nothing about the quality of the digitization effort, the authors have neither justified nor clarified, at least according to Figure 1, their choice of stations to rescue!

Apparently most of the data came from the NOAA site. And most of the remainder from the Croatian DHMZ? In the end, most rescued data came from Czech Republic (40%) and much fewer from other Balkan countries? Table 1 not helpful, need it in English s.v.p!  By stations: 11 in Czechia, 5 in Slovakia, 3 in Serbia, 2 in Boznia and 1 or none in other Balkan countries. I make this point because after the authors claimed to have chosen stations for data rescue based on spatial/temporal analysis, it seems that in fact they rescued data primarily according to convenience of access. I do not criticize here, any other data rescue project would confront the same issues? I think however the authors have claimed an analytical approach but proceeded on much more an availability approach. If true, they should clearly inform readers! Nothing from the 'red' areas of Poland, Italy, Portugal?

Page 6 line 133 "INDECIS project represented a great opportunity to rescue all this amount 134 of non-digitized daily data by using the same data sources already scanned." So this effort only involved digitization of data sheets already scanned.  Good, valuable if well done, but misleading to claim full data rescue? This team digitized data, but data that others had arleady scanned, e.g. partially rescued?

Page 6 line 139 "secondary data sources are more prominent to keep transcription errors than original data sources". So the NOAA data were already transcribed? Transcribed but not digitized? Very confusing here, a reader can not determine at what level this work started from nor subsequent levels of digitization, cross-checking, validation, etc. Does their use of the word 'keep' here imply that prior transcriptions from the scanned data sheets had already induced errors, or that the scanned sheets somehow have a higher possibility to induce subsequent transcription errors? Reader does not understand what the authors intend.

Page 6 line 146 "an inventory of candidate climate series to be rescued was created prioritizing those stations not included in ECA&D" So the data rescue effort focused on existing WMO-

labelled stations that had data in the NOAA or DHMZ archives but had not yet found their way into the ECA%D archive. Not a criticism, but very far from from the analytical approach (identify spatial and temporal gaps in key regions) hinted at earlier. This is the hard reality of data rescue efforts, the authors should admit it up front.

Page 6 line 153: Here the authors claim 50-year digitization periods for most of their data rescue efforts but in their examples (Figures 11 to 13) for WMO station 13274 (Belgrade - and do they designate that as central European or Balkan?) they only demonstrate a 15-year backward extension tail (1920-1935) on an already-available 80-year record (1936 to 2017). Not a good visual or quantitative demonstration of the impact of this effort. No validation against other stations or other sources.

Page 7 line 194: "a second and more sophisticate (sic) layer of quality control routines must be run to detect non-systematic errors hidden in climate data for future climate analysis" Who will do this subsequent necessary step? The data as rescued here remain of limited value without subsequent QA steps?

A large number of transcription error detection and validation schemes exist, several of them described and applied to other ESSD data sets. The 'key as you see' and WMO protocols cited here do not cover all the identified source and transcription errors. This reader wishes the authors had evaluated their particular source materials more carefully, referenced other transcription methodologies (including novice/expert or two-reader techniques), and done more than simply follow WMO recipes. Or, tell us why not?

For reasons already mentioned, this reader did not find the 15-year data assessments for the Belgrade station particularly useful or convincing. As larger issue, we find very limited quality control - mostly related to transcription rather than climatology or metrology - and almost no validation. Does the ECA&D data product, as amended with these newly-rescued data, now produce better records of extremes (flood or drought) across Europe? Does ECA&D now offer a better fit to re-analyses? This reader did not find evidence, beyond number of daily staton data digitized or efforts spent in digitization, that these data made any useful contribution. One presumes the authors intended such a contribution, but they have largely failed to convince readers and future users.

---

## Author Comment (AC1) · 17 May 2019

Author's responses appear in blue in the text.

**Referee 1:**

The paper describes the digitization and examination of daily climate data (610K observations of maximum and minimum temperature, rainfall, sunshine duration, snow depth) that have been digitized in the framework of a European project and which will be included in the ECA&D data set. The paper accompanies a publication of the data on PANGEA. This falls within the scope of the journal. The paper is short (well, it is not a huge data set), focused on the digitizing process, and generally well written. However, I have a few comments that I would like to authors to address.

Many thanks to the reviewer for the evaluation of this manuscript. Your comments are really welcomed.

**General:** The English is sometimes a bit awkward - please have your manuscript checked by a native speaker. I found the project in I-DARE (https://www.idare-portal.org/), but unfortunately not in the State of Data Rescue Assessment C1 ESSDD Interactive comment Printer-friendly version Discussion paper (https://rmets.onlinelibrary.wiley.com/doi/10.1002/gdj3.56). Data on PANGEAE: I did not find a legend for the "Reference" column. What does so33 mean?

The manuscript will be thoroughly revised to address general language issues.

About data on PANGAEA, datasets are accompanied by an "INDECIS_readme.pdf" file in which the structure and metadata of datasets are detailed (available at: https://doi.pangaea.de/10.1594/PANGAEA.896957). In this document, Table 2 is referred to the list of rescued data sources in which the source ID (reference column in the dataset), the source name, the period covered, the sheet format and the source provider are described. This document will be modified to clarify that "Reference" column is related to data sources and, specifically to the source ID of each data source (Table 2). Thanks to the reviewer to aware us on this.

**Abstract:** Please mention somewhere the time period covered. Abstract: Can you say something on metadata in the abstract? The systematic collection with the form given in the paper is worth mentioning.

The period covered by rescued stations and the metadata collection will be specifically included in the abstract.

**Introduction:** Perhaps the paper could cite Thorne et al. (BAMS, 2017) as a vision where all global terrestrial data set eventually could end up.

The publication carried out by Thorne et al., 2017 is worth to be mentioned in the introduction section to highlight the effort of having a global set of data holdings to provide better climate research, analysis and predictions. Many thanks for this recommendation.

**L. 151:** No pressure measurements were digitized, although they appear on the form. It is not clear whether this has already been done (it is somehow implied, but not explicit). Pressure is an important variable. Even though it is not stated in the manuscript as a focus variable, it would still have been good to collect it along with the other data.

Atmospheric pressure measurements which appear in data sources were already digitized under the UERRA project. In section 2.2 we stated: "Synoptic station-based observations of atmospheric pressure, air temperature, wind speed and wind direction were already digitized at hourly scale

under the UERRA project, but many other meteorological observations remained undigitized at daily scale." This sentence will be rewritten to clarify which variables remained undigitized at daily scale (Tmax, Tmin, precipitation, sunshine duration and snow depth). Atmospheric pressure is only available at hourly scale on data sources, thus, it was already digitized under UERRA project.

The abstract mentions "preliminary quality control". I find this very useful. However, in my view this is not sufficient for calculating extreme indices. The QC is focused only on digitising errors, not on many other possible error sources. Also, no break-point detection is applied. However, extreme indices may be sensitive to outliers as well as to biases. The authors are clearly aware of that and mention it on lines 301 and following, but then they still go on. I know that this is only for illustration of what the data could deliver, what they add to the already existing material. But perhaps this should be phrased even more carefully.

The calculation of extreme indices by using raw data is used only as an exercise to illustrate the importance of data rescue efforts for climate research. The reviewer is right in noting that a more thorough QC should be performed. We know that solid conclusions cannot be extracted from these results and it is mentioned in the manuscript. The QC applied in this study is simply to ensure the digitization procedure was carried out properly, without typing errors. Once all raw data are in the ECA&D database, more robust QC routines will be run to identify potential errors hidden in the time series (QC tests performed by ECA&D are available at: https://www.ecad.eu//documents/atbd.pdf). All data contained in ECA&D are submitted to these QC procedures to keep data reliability and consistency. Nevertheless, we will aware to the reader more carefully about the need to apply more rigorous QC tests and homogenisation techniques before any climate analysis (e.g. computation of extreme indices). A few lines mentioning what explained above will be included along section 2.5 "Computation of climate extreme indices" and section 3.3 "Preliminary assessment of rescued data" to make it clearer.

Table 1: It would be good to have more information such as number of stations, provenance (repository/archive), perhaps number of pages/images.

Table 1 will be modified accordingly.

**Referee 2:**

Regardless of its political wrappings (GFCS, WMO, etc.) does this data product add substantial reliable value to European climate data? If so, how can users verify? What have the authors accomplished by their efforts that make this data rescue effort notable and that qualifies this product for publication in ESSD? Clear answers do not emerge from the present manuscript.

Many thanks for this thorough review of the manuscript.

Regarding the first comment, a new map will be included to highlight the importance on gathering new data over Europe, especially during the first half of the 20th century. The map will show number of available stations in the early-20th century across Europe (the period which DARE data starts) to see the density of European station network. The aim of this map is to give readers an idea about the need of such DARE efforts for that period to improve data coverage over Europe. We think that this manuscript deserves to be published in ESSD journal because it reports a non-negligible amount of new observed daily data (for various relevant climate variables and different time periods) making them available to the scientific community in order to enhance the accuracy of current climate products. It resulted one year of hard work knowing the time consuming tasks derived from DARE activities. Unfortunately, there are not too many projects that include DARE activities, which are absolutely essential to perform reliable climate assessments and products. Relevant impacts of DARE efforts will be added in the "Summary and conclusions" section to highlight the importance on gathering new observed data from this kind of contributions.

Page 5 line 100: Authors introduce the acronym "DARE" which this reviewer understands as shorthand for DAta REscue but we get no definition. The authors use this term several times throughout the manuscript so we need a definition.

Thanks to alert us on this. The DARE definition provided by WMO (2016) will be included in the Introduction section:

"Climate data rescue involves organizing and preserving climate data at risk of being lost due to deterioration, destruction, neglect, technical obsolescence or simple dispersion of climate data assets over time. Non-digitized data are at risk, owing to the vulnerability of the original paper record. Date rescue includes: organizing and imaging paper, microfilm and microfiche records; keying numerical and textual data and digitizing stripchart data into a usable format; and archiving the data, metadata and the quality-control outcomes and procedures."

Figure 1 highlights two potential data deficiencies: absence of stations/data (beige background color) and existence of stations but data not downloadable (red dot). But the narrative talks about spatial data coverage and temporal data gaps, while the rescue products refer to undigitized data. In fact Spain - with relatively weak spatial distribution at least according to Fig 1 - and even France have substantial spatial data coverage problems but apparently nothing to rescue? But didn't France already address their data coverage issues through their SCOPE climate project, e.g. https://doi.org/10.5194/essd-11-241-2019? So if this study then focused on the red dot regions, e.g. where data exist but either not downloadable (according to caption in Figure 1) or not digitized (according to narrative) - and the reader gets no overt explanation of difference if any between not downloadable and not digitized - then northern Europe particularly Poland and Czechia seem most 'red'. If the Balkans represent a "a key region for data rescue missions" (page 5 line 113) that designation must include additional factors such as relative ease of access to non-digital data? Saying nothing about the quality of the digitization effort, the authors have neither justified nor clarified, at least according to Figure 1, their choice of stations to rescue! Apparently most of the data came from the NOAA site. And most of the remainder from the Croatian DHMZ? In the end, most rescued data came from Czech Republic (40%) and much fewer from other Balkan countries?

Thanks to alert us about the need to describe Figure 1 accurately. The map shows the spatial distribution of stations included in ECA&D. Daily station-based data from ECA&D are divided in downloadable (green points) and non-downloadable (red points) data. This indicates that the daily data for the non-downloadable stations are not available from ECA&D website, but can be available from the provider data itself. All data (downloadable and non-downloadable) are in digital format in ECA&D and the distinction is only related to data policies to make data publicly available (downloadable) or not (non-downloadable) through the ECA&D website. Portions of the map without red or green points mean absence of stations. Then, this study focuses on regions with lower density of stations (less presence of red/green points in figure 1). For this reason the narrative of the manuscript focuses on spatial data coverage and temporal data gaps to identify data-sparse regions/periods across Europe. Section 2.1 will be extended to make clearer the description of Figure 1.

Data rescue activities were mainly focused on Balkan region not only due to the low density of stations identified in this region according to figure 1, but also for the limitations related to data sharing derived from strict data policies. An opportunity raised to rescue data for the Balkans, but no such opportunity existed for other European data-sparse countries/regions. The Balkans was designed as key region not for the total amount of rescued data (lower than digitized for Central Europe), but for the low density of stations identified combined with the opportunity to develop DARE efforts taking into account the strict data policies that exist in such region. A clarification of the text is in need about the criteria of selection of stations for DARE purposes.

Table 1 not helpful, need it in English s.v.p! By stations: 11 in Czechia, 5 in Slovakia, 3 in Serbia, 2 in Boznia and 1 or none in other Balkan countries. I make this point because after the authors claimed to have chosen stations for data rescue based on spatial/temporal analysis, it seems that in fact they rescued data primarily according to convenience of access. I do not criticize here, any other data rescue project would confront the same issues? I think however the authors have claimed an analytical approach but proceeded on much more an availability approach. If true, they should clearly inform readers! Nothing from the 'red' areas of Poland, Italy, Portugal?

Data sources listed in Table 1 will remain with the original language, but English translations will be provided in the same table too. According to Table 4, nine stations were digitized for the Balkans (41.5% of rescued data) and 16 stations for Central Europe (58.5% of rescued data). As explained above, main focus for DARE was the Balkans due to rescued data have a greater impact in data-poor regions (such as in the Balkans) than in other data-rich regions (e.g. Germany). It is clear that there are many other areas where data rescue is required, but time and resources were too limited. At the same time, the availability of data sources is important as seen in the digitization of Central Europe. Main focus was the Balkans, but we used the opportunity to digitize also stations for Central Europe by using the already scanned data sources. Thanks for identifying this, we will add this point in the manuscript.

Page 6 line 133 "INDECIS project represented a great opportunity to rescue all this amount 134 of non-digitized daily data by using the same data sources already scanned." So this effort only involved digitization of data sheets already scanned. Good, valuable if well done, but misleading to claim full data rescue? This team digitized data, but data that others had arleady scanned, e.g. partially rescued?

In this study, original data sources obtained through the Croatian Met-Service were organized, scanned, digitized, QC'ed of digitization and data and metadata were archived. Data sources obtained through CDMP/NOAA and used in this work were digitized, QC'ed of digitization and data and metadata were also archived. According to WMO (2016), data rescue includes: "Organizing and imaging paper, microfilm and microfiche records; keying numerical and textual data and digitizing stripchart data into a usable format; and archiving the data, metadata and the

quality-control outcomes and procedures." We know that all tasks involved in DARE were not carried out step by step in this study, but already scanned data sources were used to be more efficient. If some data sources were already scanned by others (CDMP/NOAA in this case), then why these data sources should be imaged again. Actually, authors are not worried whether DARE activities are fully or partially completed, but the final goal is to make observed records available to the scientific community.

Page 6 line 139 "secondary data sources are more prominent to keep transcription errors than original data sources". So the NOAA data were already transcribed? Transcribed but not digitized? Very confusing here, a reader can not determine at what level this work started from nor subsequent levels of digitization, cross-checking, validation, etc. Does their use of the word 'keep' here imply that prior transcriptions from the scanned data sheets had already induced errors, or that the scanned sheets somehow have a higher possibility to induce subsequent transcription errors? Reader does not understand what the authors intend.

Thanks for identifying this. This statement needs to be clarified in the text. The word "keep" is not well used in this sentence. Authors tried to explain that the majority of data sources from CDMP are secondary, meaning that they are collations or summaries of observations that have been prepared in a central location. Unfortunately, secondary data sources are more prone to transcription errors than original series, since they have been transferred from the original reading. This could be traduced in low quality scans, hard to read pages, or even out of order pages, which can easily lead to digitizing errors, especially with handwritten observations. The text will be clarified accordingly.

Page 6 line 146 "an inventory of candidate climate series to be rescued was created prioritizing those stations not included in ECA&D" So the data rescue effort focused on existing WMO-labelled stations that had data in the NOAA or DHMZ archives but had not yet found their way into the ECA%D archive. Not a criticism, but very far from from the analytical approach (identify spatial and temporal gaps in key regions) hinted at earlier. This is the hard reality of data rescue efforts, the authors should admit it up front.

Thanks to alert us that this needs clarification in the text. According to page 6 line 145: "Once data sources were thoroughly inspected, the digitization plan was designed taking into account the spatial-temporal data gaps previously found in ECA&D dataset. Thus, an inventory of candidate climate series to be rescued was created prioritizing those stations not included in ECA&D in order to increase climate data spatial coverage across Europe." Authors referred that the inventory of candidate climate series to be rescued was designed taking into account data gaps previously found in ECA&D dataset (eastern Europe, Balkan region, Mediterranean basin and Central Europe). However, it is true that availability of data and limitations in resources are essential to design a digitization plan. This statement will be added in the manuscript.

Page 6 line 153: Here the authors claim 50-year digitization periods for most of their data rescue efforts but in their examples (Figures 11 to 13) for WMO station 13274 (Belgrade - and do they designate that as central European or Balkan?) they only demonstrate a 15-year backward extension tail (1920-1935) on an already-available 80-year record (1936 to 2017). Not a good visual or quantitative demonstration of the impact of this effort. No validation against other stations or other sources.

The sentence written in page 6 line 153 specifically says: "Rescued periods were variable across time covering the period 1949-2012 for the climate series located in the Balkans region and the period 1917-1968 for climate series in Central Europe." In this case, authors did not attempt to say that all stations cover the entire period, but rescued periods were variable across time comprising the aforementioned periods. In addition, table 2 shows the exact digitizing period for each station. This statement will be clarified in the text to avoid confusions. Otherwise, the

selection of Belgrade station to illustrate DARE efforts was taking into account that 15 new digitized years in the early-20[th] century are really important in this particular region. An extension of 15 years allowed to have almost 100 years long time-series in Belgrade, which surely has a positive impact on climate assessments accuracy once QC and homogeneity testing are carried out. The validation of rescued data against other stations will be undertaken during the QC procedure run by ECA&D, after ingesting raw data rescued in this study.

Page 7 line 194: "a second and more sophisticate (sic) layer of quality control routines must be run to detect non-systematic errors hidden in climate data for future climate analysis" Who will do this subsequent necessary step? The data as rescued here remain of limited value without subsequent QA steps? A large number of transcription error detection and validation schemes exist, several of them described and applied to other ESSD data sets. The 'key as you see' and WMO protocols cited here do not cover all the identified source and transcription errors. This reader wishes the authors had evaluated their particular source materials more carefully, referenced other transcription methodologies (including novice/expert or two-reader techniques), and done more than simply follow WMO recipes. Or, tell us why not? For reasons already mentioned, this reader did not find the 15-year data assessments for the Belgrade station particularly useful or convincing. As larger issue, we find very limited quality control - mostly related to transcription rather than climatology or metrology - and almost no validation. Does the ECA&D data product, as amended with these newly-rescued data, now produce better records of extremes (flood or drought) across Europe? Does ECA&D now offer a better fit to re-analyses? This reader did not find evidence, beyond number of daily staton data digitized or efforts spent in digitization, that these data made any useful contribution. One presumes the authors intended such a contribution, but they have largely failed to convince readers and future users.

The aim of this work is to provide digitized data without transcription errors to ECA&D. For this reason a preliminary QC of the digitization procedure was applied and described in section 2.3 and QC results are reported in section 3.2. It is a separate study to perform a QC to detect non-systematic errors hidden in climate data as well as homogeneity testing. Once rescued data are integrated to ECA&D, climate series are submitted to the second layer of QC routines described at: https://www.ecad.eu//documents/atbd.pdf. If there are QC problems with the data itself, then suspicious values will be flagged (and validated, corrected or rejected) in the database itself. Then, all climate series contained at ECA&D are evaluated by applying the same QC methods, also including the stations rescued in this study. Thus, data rescued in this study are delivered to ECA&D as raw data since any QC of climate series is applied. This will need clarification in the text to avoid confusions.

On the other hand, it is known that the double-keying technique, a suggested method of improving digitized data quality (Brönnimann et al., 2006), substantially increases the quality of the digitization when cross-checking the digitized stations typed, at least, twice. Nevertheless, the cost of the digitization process to be assumed is, at the same time, much higher. Then, we think that it is crucial to make a balance between quality of digitization and costs. In our case, budget constraints made unfeasible to employ double-keying unfortunately. Other transcription techniques such as optical character recognition (OCR) were previously tested in other DARE projects such as EURO4M and UERRA. From our experience, the time and costs associated with training the software for each data source and the post-processing time consuming to amend transcription errors made this option unfeasible (Ashcroft et al., 2018). For these reasons, we decided to apply the "key as you see" technique, also recommended by WMO (2016), which also provides high-quality rescued datasets with lower costs (as described in Ashcroft et al., 2018). This explanation will be included in section 2.3 to clarify the reasons about the digitization method selected for this study.

Authors think that it is beyond the scope of this manuscript to show the improvements of derived climate products from ECA&D such as reanalysis data as a result of the addition of data rescued in this study. However, there are examples about the positive impact of increasing observed data across Europe to fit better reanalysis products. For instance, Cornes et al., (2018) describe the construction of a new version of the E-OBS gridded observational dataset for temperature and precipitation back to 1950 across Europe. E-OBS is a station-based gridded dataset fed by ECA&D. As known, station density varies over time and regions which significantly affects the reliability of the estimation of interpolation uncertainty in the gridded fields. One of the main results derived from Cornes et al., (2018) is that the uncertainty is still underestimated in data-sparse regions. Thus, the best way to improve the reliability and accuracy of a gridded observational dataset is to provide more observational data, especially in poor-data regions and periods. We think that our study contributes to increase the station density in data-sparse regions during the early-20$^{th}$ century (among other periods), when station data availability is limited. As a consequence, we think that this contribution will surely help to enhance the reliability of such derived climate products.

Authors are convinced that the addition of 610K observations to ECA&D it is not a negligible amount of new data knowing the need of new observed data (especially in regions with strict data policies) to create more accurate climate products.